# Outlier Detection with Reinforcement Learning for Costly to Verify Data

**DOI:** 10.3390/e25060842

**Published:** 2023-05-25

**Authors:** Michiel Nijhuis, Iman van Lelyveld

**Affiliations:** 1De Nederlandsche Bank, 1000 AB Amsterdam, The Netherlands; 2Department of Finance, VU Amsterdam, 1081 HV Amsterdam, The Netherlands

**Keywords:** reinforcement learning, algorithm design, outlier detection, anomaly detection, outlier ensembles

## Abstract

Outliers are often present in data and many algorithms exist to find these outliers. Often we can verify these outliers to determine whether they are data errors or not. Unfortunately, checking such points is time-consuming and the underlying issues leading to the data error can change over time. An outlier detection approach should therefore be able to optimally use the knowledge gained from the verification of the ground truth and adjust accordingly. With advances in machine learning, this can be achieved by applying reinforcement learning on a statistical outlier detection approach. The approach uses an ensemble of proven outlier detection methods in combination with a reinforcement learning approach to tune the coefficients of the ensemble with every additional bit of data. The performance and the applicability of the reinforcement learning outlier detection approach are illustrated using granular data reported by Dutch insurers and pension funds under the Solvency II and FTK frameworks. The application shows that outliers can be identified by the ensemble learner. Moreover, applying the reinforcement learner on top of the ensemble model can further improve the results by optimising the coefficients of the ensemble learner.

## 1. Introduction

In many datasets, errors are present. To fruitfully analyse or run machine learning algorithms on a dataset, the quality of the data should be adequate. One of the ways to increase data quality is to remove data errors from the data. These data errors could manifest themselves as outliers or anomalies, data points that are ‘far’ from ‘normal’. However, if these outliers are not data errors, they can be informative as they show behaviour that is very different from what is happening in the rest of the data. In many cases, data can be verified although verification often still includes manual and hence costly actions. An example of this is firms’ financial reporting: If companies report their financial accounts, business rules can be applied to see if the numbers add up. For instance, assets should equal liabilities. If, however, for example the turnover in a certain market is significantly higher than before, something could have gone wrong in the collection of the data. Alternatively, the company could have increased its focus on that particular market and consequently increased sales. If it is a data error, it should be corrected while if it is a true trend it probably will have to be explained. If we would know beforehand which of these two options holds, then data checks could be further automated. With the increase in the volume of data that companies need to report, accurate outlier detection becomes ever more important.

Removing outliers from data is important in many situations across almost all disciplines. In finance, for instance, outlier detection is used to detect credit card fraud [1]. In the medical field, outlier detection can be used to assist the diagnosis several diseases [2]. In the environmental field outlier detection can be used to identify pollution [3]. In manufacturing, outlier detection can provide warnings of faults in the production process [4]. In many of these cases there is only limited labelled data available and generating new labelled data would be costly. An outlier detection method for these kinds of applications should therefore be able to perform well even with limited labelled data and be able to improve its performance which each additional labelled data point.

In the literature, many approaches have been proposed to find outliers in data. One of the main approaches is to distinguish between normal, regular behaviour and ‘outlier’ behaviour. To establish what non-regular behaviour looks like several approaches have been proposed, many of them implemented in Python [5]. First of all, simple statistical methods can be used to determine if a certain point is too far away from the bulk of the data [6]. Using the now currently often employed expert rules to find these outliers would no longer be a cost-effective way to find these outliers [7]. In addition, there have been many advances recently compared to established statistical approaches. Distance and clustering-based metrics are employed to indicate how far points are separated [8,9], density methods have been used to spot areas in the data with a divergence in density [10,11], graph-based approaches focus on connections to other points that are out of the ordinary [12,13] and machine learning approaches categorise outliers by assessing how difficult it is to predict a data point [14,15] which can be enhanced by the application of reinforcement learning [16,17].

These approaches often just indicate whether a certain point is an outlier or not. If information is available about which points are verified real outliers then this ground truth can be used to tune the parameters of the outlier detection algorithm. This hyperparameter tuning can lead to significantly better predictions depending on the algorithm applied [18]. For the tuning of the parameters, the ground truth of many points needs to be known and in many cases this is not the case. Therefore, one wants to use every additional bit of additional available information to tune the outlier detection. This can be performed using a reinforcement learning approach in combination with an outlier detection algorithm. A simple approach is one in which the ranking of the outliers is adjusted retroactively [19,20]. This can be improved using a true reinforcement learning approach [21]. If verification can be obtained from subsequent time steps then a time series approach can be employed [22]. A reinforcement learning application can also be trained on similar data and applied to a new dataset (i.e., transfer learning. See [23]). These approaches, however, do not make full use of the specific data to generate a better performing outlier detection. This is inefficient, given that it is costly to evaluate the data. We have therefore developed a new reinforcement learning outlier detection approach that can make use of the additional information in verified outliers.

In this paper, we will discuss how we combined a basic outlier detection method with a reinforcement learning approach to ensure these statistical methods perform with higher accuracy. First, we discuss the basic outlier detection algorithm in Section 2. Then, in Section 3, the reinforcement learning approach is discussed. Hereafter, in Section 4 we apply the reinforcement learning outlier detection to real data to evaluate the performance. The paper finishes with the conclusions in Section 5.

## 2. Outlier Detection

An outlier detection algorithm should be able to handle many different kinds of outliers. The main types of outliers are values that are either too high or too low with respect to their peers and values with a higher density compared to their peers (for instance, if exactly the same information is given continuously). Detecting these two types of outliers with a single algorithm is challenging. For this reason, we propose an outlier detection based on an ensemble of different outlier algorithms. An ensemble model combines multiple individual models. In this paper, the ensemble consists of three different methods for outlier detection: an interquartile range, nearest neighbour distance and a local outlier factor. First, we will briefly discuss these three methods in turn.

### 2.1. Interquartile Range

The interquartile range is a basic measure to identify outliers [24]. To calculate the range the following formula is applied: (1)$$iqr_{i}=\frac{x_{i}-f_{50}}{f_{75}-f_{25}}$$
where $x_{i}$ is the sample from the data for which the interquartile range iqr is calculated, $f_{75}$ is the 75th percentile value from the complete set of data *f* and $f_{25}$ the 25th percentile value and $f_{50}$ is the median value. The interquartile range thus gives an increasing score to a point as it gets further away from the median. By making this score dependent on the range between the 25th and 75th percentile values the score would be lower for data with a large variance compared to data with a small variance.

### 2.2. k-Nearest Neighbour Distance

The second metric which we apply, is the average distance to the k-nearest neighbours [25]. The score for the k-nearest neighbours is calculated as follows: (2)$$nn_{i}^{k}=\frac{1}{\bar{nn^{k}}}\sum_{n\in k}\left|x_{i}-x_{n}\right|$$
where $nn_{i}^{k}$ is the nearest neighbour score for point *i* when we assess *k* nearest neighbours, $x_{i}-x_{n}$ the distance between the point for which we calculate the nearest neighbour score $x_{i}$ and one of *k* its nearest neighbours $x_{n}$ and $\bar{nn^{k}}$ is the average nearest neighbour score for the *k* nearest neighbours. Where the interquartile range penalises points based on their distance to just the median point, the nearest neighbour score penalises points based on their distance to all other points.

### 2.3. Local Outlier Factor

The last metric which we include in the outlier detection ensemble is the local outlier factor [26], calculated as follows: (3)$$lof_{i}^{k}=\frac{\sum_{j\in N_{x}^{k}}l_{j}^{k}}{l_{i}^{k}\left|N_{i}^{k}\right|}$$
where $lof_{i}$ is the local outlier factor, $l_{i}$ is the local reachability distance *i*, $N_{i}^{k}$ is the set of k-nearest neighbours of *i*. The local reachability distance in turn is defined as: (4)$$l_{i}=\frac{\left|N_{i}^{k}\right|}{\sum_{j\in N_{i}^{k}}\max\left[d_{j}^{k},d_{i,j}\right]}$$
where $d_{j}^{k}$ is the distance to the *k*th nearest neighbour of *j* and $d_{i,j}$ is the distance between *i* and *j*. The local outlier factor can be seen as a measure comparing the local density of a point to the local density of the points around it. If the local density is higher (lower) than the density of the points around it, a local outlier factor higher (lower) than one will be the result. A local outlier factor lower than one also indicates possible outliers. If some points show *exactly* the same behaviour it can also be a sign that these values are outliers, for instance in the case of default values. For the other two methods an increase in the likelihood of being an outlier is connected to an increasing score. We therefore have to adjust the local outlier factor to have a score comparable to the other methods in the ensemble. A score of zero means no outlier and the higher the score, the more likely the point is an outlier. We achieve this adjustment using the following formula: (5)$$lof_{i,adjusted}^{k}=\left\{\begin{array}{cc} \frac{1}{lof_{i}^{k}}-1, & lof_{i}^{k}\leq 1\\ lof_{i}^{k}-1, & lof_{i}^{k}>1 \end{array}\right.$$

### 2.4. Ensemble Model

The three measures we discussed above form the basis of the ensemble model. To understand their interaction, we plotted the score for each of the measures in Figure 1. The horizontal axis indicates the value of the points. This value is the one-dimensional parameter we use for the detection of outliers. The score for each of these points is thus calculated solely based on their position on the *x*-axis.

From the figure, we can see that the three methods complement each other well for the hypothetical distribution of points shown. In this case, the endpoints are the clearest outliers as they score high for each method. The points in the middle with a higher density score low on the nearest neighbour metric as the distances to the nearest points is limited while they score high on the local outlier factor as their local density is significantly different compared to the surrounding points.

For these three metrics, the k-nearest neighbours and the local outlier factor can both be calculated for different magnitudes of *k*. These different values of *k* can lead to completely different outcomes of the scores, while the number of points would also have an effect on the results for a certain value of *k*. In the ensemble model, we therefore, employ both the nearest neighbour and the local outlier factor approach with two values of *k*, a low and a high value. These values are optimised depending on the number of points to which the outlier detection is applied. The high value is equal to $\left\lceil \frac{\sqrt{n}}{2}\right\rceil$, where *n* is the number of points to which the outlier detection is applied. The low value is equal to $\left\lceil \frac{n^{0.2}}{2}\right\rceil$. If low and high values are equal only one of them is used.

To be able to combine the different algorithms included in the ensemble, multiple methods can be employed. Commonly, a weighted average ensemble approach is used for the combination of different models within an ensemble [27]. This uses weighted soft voting in which the probabilities of the different classes (outlier or inlier) are summed using weights. The models employed in our ensemble have very different distributions when it comes to separating the outliers. Using weighted soft voting directly to these values could have unintended consequences. First of all, the different models should produce results within the same range. In order to do this, several different methods can be employed. A min-max scalar is most often used to scale the data to a predefined range. Another options is rank-scaling. Rank scaling ranks the points according to the score and subsequently scales this rank to the predefined range. This allows for intensifying small differences between points. The downside of this is that the information of the distance between the points is lost. The downside of a min-max scalar is its susceptibility to extreme outliers. We therefore employ a combination of the rank based and min-max scaling. The scaling is given by the following function: (6)$$p_{x,new}=\frac{r\left(p_{x}\right)}{\bar{p}}+\frac{p_{x}-\min p_{x}}{\max p_{x}-\min p_{x}}$$
where $p_{x,new}$ is the new value for point $p_{x}$, $\bar{p}$ the cardinality of the set of points *p* and r() indicates a ranking function. The ranking function ranks the points from a certain inlier (1) to a certain outlier ($\bar{p}$). In the $\frac{r\left(p_{x}\right)}{\bar{p}}$ is the rank based scaling, the $\frac{p_{x}-\min p_{x}}{\max p_{x}-\min p_{x}}$ is the min-max scaling.

In the ensemble, all the scores are combined into a single outlier score. Depending on the actual outliers in the data, the different outlier detection algorithms should be weighted differently. Dynamically adjusting these weights has been shown to increase the accuracy of ensemble models [28]. The weighted soft voting to come to a single ensemble score is implemented as follows:(7)$$o_{i}=\frac{1}{c_{nn}^{k,high}+c_{nn}^{k,low}+c_{lof}^{k,high}+c_{lof}^{k,low}+c_{iqr}}\cdot \alpha$$
(8)$$\alpha =c_{nn}^{k,high}\cdot nn_{i}^{k,high}+c_{nn}^{k,low}\cdot nn_{i}^{k,low}+c_{lof}^{k,high}\cdot loc_{i}^{k,high}+c_{lof}^{k,low}\cdot loc_{i}^{k,low}+c_{iqr}\cdot iqr_{i}$$
where $c_{nn}^{k,high}$, $c_{nn}^{k,low}$, $c_{lof}^{k,high}$, $c_{lof}^{k,low}$, $c_{iqr}$ are the coefficients for the different outlier algorithms and $o_{i}$ is the final outlier score. The choice of these coefficients can have a large effect on the performance of the outlier detection and multiple ways exist to combine the scores [29]. We determine these coefficients depending on the results from the reinforcement learner and use them to tune performance. We will explain how the reinforcement learning algorithm adjusts these coefficients in the next section.

After we score the outliers, the next step in the process is setting the threshold which determines when a point can be considered an outlier. Depending on the data, this point can vary. Moreover, the algorithms applied during the outlier detection are in and of themselves prone to outliers [30] as the scores are normalised not purely based on rank. The weighing of the different methods in the ensemble could therefore be skewed by large outliers from a single metric which can obfuscate outliers within the data. There are multiple ways of dealing with this issue. Completely converting to a rank-based normalisation would be one feasible solution. However, the information of the actual scores of the metrics would then be lost. Another way in which this issue can be resolved, is by looking at the distribution of the resulting scores for each metric and making an adjustment to the scores of the individual metrics based on this distribution. Note that the distributions of the individual metrics can also shift over time.

We employ an iterative approach to setting the threshold: we remove the top outliers from the data and then reapply the outlier detection algorithm, continuing until a stopping criterion is met. In this way, we reduce the sensitivity of the outlier detection to outliers while we leave the data and the scores unadjusted. This leaves the choice of how many of the top outliers should be removed from the data during each of the iterations and the choice of the stopping criterion open. As the iterations are performed to reduce the impact of large outliers on the score of a single metric, the iterations should at least continue until there are no more very large outliers. We make the stopping criterion dependent on the distribution of the final outlier score by basing the stopping criterion on the distribution of the final calculated score. We remove the top *n* outliers until (1) this distribution becomes stable and (2) no large outliers remain.

As the distribution of the scores can vary, we choose a non-parametric approach to determine the stability of the distribution of the scores. We calculate the differences in certain percentile values of the scores between two iterations and if these are stable, the iterative process is stopped. This can be expressed as follows: (9)$$R_{stop}>\frac{|o_{t}^{99.9}-o_{t-1}^{99.9}\left|+\right|o_{t}^{99.5}-o_{t-1}^{99.5}|}{|o_{t}^{99.9}+o_{t}^{99.5}|}$$
where $R_{stop}$ is the stopping threshold, $o_{t}^{99.9}$ the 99.9th percentile value of the set of final outlier scores *o* at iteration *t*. We need to set the stopping threshold depending on the data to which the outlier detection is applied. All the data which will have outlier scores $o_{i}$ above a threshold value of $O_{threshold}$ will be considered an outlier in all of the iterations.

## 3. Reinforcement Learning

A reinforcement learning method can help find the optimal coefficients for the outlier detection ensemble over time and in this section, we describe such an algorithm. First of all, a general overview of the reinforcement learning algorithm is given, followed by a closer look at the model the agent applies and the policy adjustments the outlier detection makes.

### 3.1. Overview

When running the algorithm we do not know whether the points classified as outliers are actually outliers. Checking whether an outlier is also a true outlier is usually a costly and time-consuming process. Therefore it would be desirable if the quality of outliers improves as more true outliers have been identified. To achieve this, we propose to use a reinforcement learning approach so that the coefficients within the outlier detection algorithm can be learned over time. This not only makes the outlier detection less sensitive to the initial values of the coefficients but also allows for the scaling of the coefficients as the outliers in the data change. If we detect and subsequently verify outliers in the data, it is likely that action will be taken to reduce the presence of these types of outliers. This could lead to a shift in the types of outliers present in the data and in this way to a shift in the importance of the different algorithms within the outlier detection ensemble.

In a standard reinforcement learning algorithm, an agent takes certain actions which affect the environment and based on the change in the environment the agent is given a (possibly negative) reward [31]. The agent in the reinforcement learning algorithm is a set of decision rules. The policy the agent used to take action is adjusted based on the reward that it receives. Next to the reward, the agent also receives an updated version of the state of the environment on which it should apply its policy to generate its next action. For the outlier detection algorithm, the policy of the agent is setting the coefficients of the outlier detection algorithm. When the agent has set the coefficients, outliers can be flagged. We can subsequently assess whether these outliers are true outliers or not by manually verifying the flagged outliers. We will use this information to give a reward to the agent. If the agent receives the information about which of the points it classified as outliers, it can also estimate what will happen if the policy is adjusted. The agent can thus use a model for the determination of what the next step will be. For the outlier detection algorithm, the use of a model is essential as the sample efficiency is far higher for model-based reinforcement learning and the cost of evaluating outliers is high.

We have made a schematic overview of the reinforcement learning employed for the outlier detection in Figure 2. The main aspects of reinforcement learning are the model the agent uses to help it find the optimal policy (i.e., a set of coefficients for the outlier detection) and how the policy is adjusted from one step to another. We will discuss these in more detail below.

### 3.2. The Agent’s Model

The agent uses a model to search for its optimal policy which should accurately approximate the actual feedback the agent would get from running the outlier detection algorithm. The simplest model would just be a replication of the outlier detection algorithm in a way that resembles self imitation learning [32]. The agent can then keep track of previous times the outlier detection algorithm was run and the result of each of the lines which were evaluated. In this way, the agent will build up more knowledge about the possible results of changes to the outlier detection coefficients as time goes by. The agent will not have any information about the outliers in this period, nor will it have any information about points that were not previously classified as outliers as these would not have been evaluated. This agent model should thus be capable of handling points where it is unknown whether or not a point will be an outlier. As time goes by the nature of the outliers might change as the reasons for the outliers could get resolved. This would thus require the agent model to adapt to new situations. In order to achieve this, we do not base the agent model on the full set of historical data, but just on a recent subset. With these two points in mind, the agent model will be an adjusted version of the outlier detection algorithm. The basic algorithm is the same, only for the evaluation of points the following formula is used:(10)$$s_{a}=\left\{\begin{array}{cc} 1, & a\in A^{r}\mathrm{and}o_{a}>O_{threshold}\\ 0, & a\notin A^{r}\mathrm{and}o_{a}<=O_{threshold}\\ -1, & a\in A^{r}\mathrm{and}o_{a}<=O_{threshold}\\ s_{exp}, & \mathrm{otherwise} \end{array}\right.$$
where $s_{a}$ is the score for point *a* in the set of all of the data points *A*, the set $A^{r}$ is the subset of the agent *A* for which a solution is known, $s_{exp}$ is the exploration parameter. The agent gets a payoff depending on the summation of the scores of the data points i.e., $\sum s_{a}$. The agent tries to maximise the sum of the outlier score of all the evaluated points. Data points which are classified as outliers which are actually inliers are explicitly penalised, whereas outliers that are missed are implicitly penalised. The score of these points would be 0 (with a possible maximum of 1). The exploration parameter indicates how many unknown points should be penalised. If the parameter has a high value, data points without a label will contribute significantly to the payoff. Many unknown points will therefore be included in the next iteration of the outlier detection, leading to an increase in the number of outliers that will have to be evaluated. We will set the exploration parameter based on the precision of the last outlier detection. A low precision indicates that either the wrong points are selected or that too many points are selected. In this case, we should therefore reduce the value of the exploration parameter, while if the precision becomes too high, the likelihood that outliers are missed increases and we should employ a higher exploration parameter. We therefore estimate the exploration parameter with the following formula:(11)$$s_{exp}={\left(\frac{\sum_{a\in A^{r,last}}\left\{\begin{array}{cc} 1, & o_{a,last}>O_{threshold}\\ 0, & o_{a,last}<O_{threshold} \end{array}\right.}{|A^{r}|}\right)}^{3}$$
where the subscript ${}_{last}$ is used to indicate the last time the outlier detection ran. The exponent 3 is used to reduce the exploration parameter to a lower level, as a high precision would be necessary to be able to validate all of the suggested outliers.

The environment in which the agent is acting consists of a subset of all the data the outlier detection has encountered so far. The data in the environment of the agent are given a stochastic decay where each point has a chance of disappearing from the data the agent model uses. We use a stochastic decay for the data as it is unknown whether a certain in- or outlier would still be important in the future. The stochastic decay prevents dropping all data with a certain timestamp all at once. In this way we can prevent oscillations within the algorithm as the coefficients of the model will at a certain time no longer be based on a period with a certain type of outliers. However, it does allow these outliers to resurface again in the future. We choose to use a truncated Gaussian distribution for the decay, to ensure all points are replaced at some point and all points are used at least once in setting the policy. The parameters of the truncated Gaussian should depend on the problem; if types of outliers are expected to disappear quickly (slowly) from the data after their discovery a low (high) mean and standard deviation should be employed. When we set the parameters we should also take into account that the computational power required by the agent model increases as we increase the mean of the truncated Gaussian distribution.

The chance that a certain data point will present in the environment of the agent can be expressed by the following formula: (12)$$P\left(a\right)=\sum_{t\in T}\alpha |N(\mu ,\sigma ,l)|$$
where $P_{a}$ is the probability that data point *a* will be in the environment, *T* the set of time steps *t* since the data point was first in the dataset, α the decay factor and N(μ,ρ,l) a truncated normal variable with mean μ and standard deviation σ, cut-off at values higher than l∗σ from the mean.

### 3.3. Policy Adjustment

The agent uses the model described to search for the optimal parameters for the outlier detection algorithm. The search uses a genetic algorithm for optimising the coefficients of the outlier detection algorithm. This approach allows for optimisation of coefficients in a non-linear search problem with a search space too large to fully explore in a reasonable amount of time. The genetic algorithm thus tries to minimise the combined score ($\sum_{a\in A}s_{a}$) of all the points in the agents’ set (*A*). The genetic algorithm represents a solution as a vector of the coefficients. The crossover between two vectors is performed by switching a single value between two vectors on the same position and the mutation is accomplished by a random increase/decrease of a single value of the coefficients vector. Finally, the selection is performed based on a two-stage tournament without replacement.

After the agent has determined the optimal policy based on the genetic algorithm and its internal model, a new policy is set. The amount of information that comes from a single iteration should be limited, to avoid a too specific sets of parameters. An information criterion can be used to identify the amount of specificity in a algorithm and steer reinforcement learning [33]. However, instead of applying this approach a more straightforward approach is taken, as the update speed of the reinforcement learning is very low. We obtain the new coefficients by taking a weighted average of the old coefficients and the optimal coefficients:(13)$$c_{new}=\frac{w_{c}c_{old}+c^{*}}{w_{c}+1}$$
where $c_{new}$ is the new coefficient, $c_{old}$ the old one, $c^{*}$ the optimal one and $w_{c}$ a weighting factor. The weighting factor can be chosen depending on the problem at hand and usually ranges between 3 and 6.

## 4. Application to Asset Data

To test whether the outlier detection algorithm performs well we apply it to the financial reporting data from Dutch pension funds and insurers. These entities are required to provide the regulator with, amongst others, a detailed overview of their assets and liabilities on a quarterly basis. Based on this data, the regulator can assess whether the entities are not taking any undue risks and have sufficient assets to meet their obligations. One of the most important data points in the data which should be accurate is the value of the assets. The valuation of the individual assets can vary significantly over time and it is therefore not straightforward to pick out the variations in asset pricing which might be data errors. To see how the outlier detection method we propose can be applied to these data we first take a closer look at the data. This is followed by a discussion of how the outlier detection can be applied to this specific dataset and, finally, the section concludes with the results of the outlier detection.

### 4.1. Solvency II and FTK Data

In this section, we take a closer look at the data reported by Dutch pension funds and insurers. The insurers’ and pension funds’ data are part of the Solvency II and FTK reporting frameworks, respectively. Since the latter is based on former framework, the data from these two sources can be combined to generate a larger dataset. The Solvency II and FTK reporting cover a wide range of accounting and risk numbers but the focus here is on the line-by-line asset data provided in tables S.06.02 (EIPOA based reporting) and K.2.08 (DNB based reporting), respectively.

For the outlier detection, we used the data from the last three quarters of 2019. The FTK data contains data from 210 pension funds with an average of 6846 unique assets per fund. The Solvency II data contains 136 insurers with an average of 234 unique assets per insurer. The sizeable difference between the number of assets of insurers and pension funds is mainly due to the difference in reporting requirements: Pension funds have to report the individual holdings of investments funds, whereas insurers can just report the overall exposure to an investment fund. As for this application we are interested in whether the value of the asset is reported correctly—and not type of exposure—we apply a uniform way of looking at the different assets. For all the assets, a price per unit is defined. This price per unit is equal to the unit stock price for equities or the value of a bond over the nominal amount for bonds. For some investments, unit prices are sometimes not available and reporting agents are then free to provide their own estimate. Therefore the change in the unit price is calculated, as this should yield comparable values. Taking the change in unit price implies that we cannot detect outliers in an initial time period but it does allow us to look at the increase in value of for instance a real estate investment fund and compare it to the change in value of similar funds. Furthermore, note that assets can be sold in one quarter and (possibly) bought back a few quarters later and that certain assets are thus not in the data for two consecutive quarters. This is the case for 25.3% of the assets.

An asset’s price change usually correlates with the price changes of other assets in the same class, country, credit rating, owner, sector or currency. As this information is also available, we use it for outlier detection. The class of assets is given by the Complementary Identification Code (CIC) and the sector is given based on the NACE subdivision. The credit rating is harmonised across the different credit rating issuers to six credit steps. Based on these variables we can generate groups of data that can be used for the outlier detection. These groups can consist of a combination of up to three of these variables as with too much differentiation the groups get too small to be meaningful.

### 4.2. Application of Outlier Detection

When we apply the outlier detection to classify the outliers into different subgroups, the additional variables class, country, sector and currency are also taken into account. The outlier detection can subsequently be applied for each of the subgroups. As it is still unknown which of the subgroups might generate a better grouping for the outlier detection, this can be included in the reinforcement learning approach. The coefficients of the different algorithms can also be seen as a set of coefficients for each of the groups. In this way the coefficients will change after it turns out that subdividing the data into a certain group is either helpful or harmful. If, after some time, it turns out that certain ways of grouping the data have a very low coefficient, these can be discarded from the data. If there are multiple groups, we need to adjust the evaluation of the ensemble outlier detection to allow for these groups. Equation (7) should be adjusted to allow for all the groups to be included giving:(14)$$o_{i}=\sum_{g\in G}\frac{1}{c_{nn}^{k,high,g}+c_{nn}^{k,low,g}+c_{lof}^{k,high,g}+c_{lof}^{k,low,g}+c_{iqr}^{g}}\cdot \alpha^{g}$$
(15)$$\alpha^{g}=c_{iqr}^{g}\cdot iqr_{i}^{g}+\sum_{a\in (high,low)}c_{nn}^{k,a,g}\cdot nn_{i}^{k,a,g}+c_{lof}^{k,a,g}\cdot loc_{i}^{k,a,g}$$
where *g* indicates a certain subgroup from the set of all the subgroups *G*. With this adjustment, we can apply the outlier detection to our data. To initialise the weights of the subgroups they are set equal to each other.

### 4.3. Results

To be able to judge the performance of the outlier detection as well as to see whether the reinforcement learning approach is working as expected, reporting experts manually checked a selection of the outliers which were found by the algorithm. The algorithm is also applied to this data, so the performance can be judged. The algorithm uses the parameters as indicated in Table 1. The results of the outlier detection for these three periods are given in this subsection. First of all, we take a closer look at whether the algorithm actually performs as expected. Then we discuss selected outliers in more detail, followed by an assessment of the reinforcement learning approach.

### 4.4. Algorithm Performance

The first set of results we are discussing determines whether the outlier detection performs as expected and whether the assumptions we made in constructing the model hold. The two main things to check are the distributions of the scores for the different methods from the ensemble and whether—after several iterations—the stopping of the algorithm performs according to Equation (9). First, we take a closer look at the distributions of the different algorithms as there are still significant outliers in the data. We plot the individual distributions of the outlier scores for the four different methods in the ensemble in Figure 3 for both the initial and the final iteration. We normalised the scores in the figure according to Equation (6).

From the figure, we can see that for some of the methods in the ensemble, very large outliers still play a role at first, even though the partly rank-based normalisation already reduced the tails of the distributions. At each subsequent iteration, the distributions converge and the influence of the few large outliers diminishes. In the final iteration, the distributions resemble each other rather closely, indicating that they can be used with the current normalisation in the ensemble.

As the outlier detection is performed iteratively, it is important to see how the stopping criterion is reached. In Figure 4 we therefore plot the change between two sequential iterations, given by Equation (9), using the entire dataset.

From the figure, we can see that for the complete dataset the distribution is stabilising as the iterations progress. In the initial distribution, a few large outlier scores push most of the scores close to zero. This effect diminishes as the iterations progress. After four iterations the distributions seem stable and the stopping criterion is reached on the fifth iteration.

### 4.5. Ensemble Performance

In this section we analyse the performance of the ensemble and compare it to the performance of other methods. To assess whether the ensemble performs adequately, we compare the ensemble to each of the three individual methods included. We also compare the results to two other methods from the PYOD package [5]. First, the cluster-based local outlier factor which is an extension of the local outlier factor to even better detect groups within the data and, second, XGBOD which is an outlier detection based on the XGBoost algorithm. For this comparison we selected the top 500 outliers according to each of the methods and calculated the precision. The results of this is given in Table 2.

From the table, it is clear that the ensemble of the three models (LOF, NN, IQR) performs better than each of the separate parts. This shows that it is an advantage to apply these methods in an ensemble rather than apply only one of the methods. From the table, we can also see that the three methods that make up the ensemble are inferior to the other two outlier detection algorithms, with the XGBoost based outlier detection performing best. The ensemble model is, however, still outperforming the XGBoost based outlier detection. This shows that the ensemble model, even without the added benefit of the reinforcement learning can perform well.

### 4.6. Outlier Selection

Next we inspect the actual selection of the outliers more closely. In order to assess whether the outlier detection is actually performing as expected, we show the results for the assets belonging to the CIC 13 category (money market government funds) as well as a table with the overall results.

First of all, we plot the results for a subset of data in Figure 5. We show the per unit value change against the score to determine whether the points with high or low per unit change values is identified correctly.

The figure shows a v-shape. The points with the lowest score have a per unit change of 5.2% but anything with a margin of around 10% can still score low. When the per unit change increases or decreases the score increases, with the highest scores for the per unit changes of around −0.8. These values indicate that a price decrease of 80% can be considered an outlier. In the middle of the v-shape, there are also some points with relatively high outlier scores. Some of these values are exactly zero and because of that have a far higher local density, resulting in a higher outlier score. The presence of the outliers on the left prevents that these points get a higher outlier score.

Another way to look at the results is to see how the outlier scores differ for a single asset. To that end, we plot the per unit change versus the score in Figure 6 for a typical asset.

From the figure, we can see that this particular asset occurs eight times in this dataset. There is about a 6% difference between most of the values and two potential outliers. For the main group of points to the right, there is even a small difference between the per unit change in value, most likely caused by differences in rounding between different reporting institutions. The outlier scores of these points are all close together and relatively low. The outlier scores for the two points on the left are significantly higher as the per unit change values can be considered outliers as they are further away from the centre. While the deviation of the per unit change of the two left points is equal, their score is (slightly) different because they have been included in different subgroups.

We can get some sense of the overall performance of the outlier detection from Table 3, in which some of the characteristics of the data are given before and after the application of the outlier detection algorithm.

In the table, we show the per unit change in value before and after the application of the outlier detection. A negative value means that the per unit price of the asset decreased. If the per unit change drops to −1 the value of the asset has become worthless (zero) in the new period. From the table, we can determine that the largest outliers have been removed from the data. The largest outlier before the application of the outlier detection is completely unrealistic and could have easily been removed by a simpler approach. Even the 99.5th percentile value is higher than would be normally feasible. The 99.5th percentile value after the application of the outlier detection is high but feasible. The maximum value is still infeasibly high and manual inspection also confirmed it was a false negative. A lower threshold (−0.02) for the outlier score would have correctly classified this point as well.

The value for the largest negative change in the data before the application of the outlier detection algorithm is listed as −1; however, the actual value is slightly higher as the taxonomy of the data does not allow zero as a value for an asset. The minimum values also show less extreme cases after the application of the outlier detection. We inspected the reduction in asset value of −0.95 and concluded it was not an outlier, but a valid data point.

Values of exactly zero can occur in the data, but are mostly due to reporting inconsistencies: the reporting institution just reports exactly the same value as the previous period. In about half of the cases, the values are given as outliers and are removed by the application of the outlier detection algorithm. Not all of these zero values are removed; they are classified as outliers depending on the prevalence of zero values in the subgroups they belong to.

We also inspected some of the potential outliers by hand to judge whether these are actual outliers or are false positives. Based on the Q3 2019 data, we identified 6283 outliers when running the algorithm. To test the performance we examined 100 of the outliers with the lowest scores which were removed during the last iteration in combination with 100 non-outliers with the highest score during the last iteration. This resulted in a precision of 73% and a recall of 76% for these 200 data points. Reducing the threshold score might slightly increase the recall and precision values. The real recall and precision would be higher as we can assume that the score for the points further away from the threshold would have been classified better. If we assume that all these points are classified correctly, the precision increases to 96% and the recall to 97%. These high values allow us to check only a few data points without having to accept large numbers of false positives/negatives.

### 4.7. Reinforcement Learning

Next to determining how well the outlier detection performed, we should also evaluate the performance of the reinforcement learner, which sets the coefficients of the outlier detection. To this end we classify the data for the outlier detection after the application of the outlier detection. Based on this classification, the reinforcement learning part is activated and the coefficients of the outlier detection algorithm are adjusted. The outlier detection can subsequently be applied to the data of the next period. To show the effects of adjusting the ensemble model, we look at the next quarter with and without the adjusted ensemble model. In Table 4 we show the statistics for the data points after the application of the outlier detection algorithm.

Looking at the table we can see a clear difference between the results of the outlier detection with the initial and with the adjusted ensemble model. Although the minimum value for both methods is the same, the percentile values and the maximum values are closer to the median value of 1.06 for the adjusted ensemble model. This indicates that the reinforcement learning part of the outlier detection is working as expected.

To further evaluate the performance of the reinforcement learning part of the outlier detection algorithm, the data points on the boundary between outliers and non-outliers are investigated. We examine 100 of the outliers with the lowest scores which were removed during the last iteration as well as 100 non-outliers with the highest score during the last iteration. We do this with and without the application of the reinforcement learning part of the algorithm. The results are shown in the bottom two rows of Table 4.

Both the recall and the precision improve when the ensemble model is adjusted through the reinforcement learning approach. With a recall and precision of roughly 75% for the initial ensemble model, the values do not differ much from the recall and precision for the Q2 data (76% and 73%), with the same ensemble model. For the adjusted ensemble model the values improve with several percentage points. This reduces the number of false positives that have to be checked by hand while finding a similar number of data errors. For the 200 points that are checked by hand for the initial ensemble model the 15 with the lowest score were all still correctly classified as well as the 7 with the highest score. Therefore, effectively, only 178 out of 200 points need to be checked. If the same number of errors is acceptable, only 152 out of the 178 points need to be checked. By applying just a single reinforcement learning step we can thus already reduce the workload by 17%. The effects after subsequent applications can be expected to be even higher as the coefficients of the ensemble model will get closer to optimal.

## 5. Conclusions

In this paper, we proposed integrating a reinforcement learning approach with statistical outlier detection algorithms. The outlier detection algorithm uses an ensemble of standard statistical outlier detection algorithms to detect different kinds of outliers. We have shown that this ensemble performs better than the individual parts. Around this ensemble we developed a reinforcement learning algorithm, to set the values for the coefficients of the ensemble model in an adaptive manner. In this way, the performance of the ensemble model can increase as data points are verified and even a small number of verification can improve the performance. This algorithm would thus be good for applications where the checking of whether a data point is an outlier is costly. The improvement over many other methods for outlier detection is that we incorporate a feedback mechanism that can improve the delineation between outliers and non-outliers.

To showcase the strength of the algorithm described in this paper, we applied the algorithm to the asset data as reported under the Solvency II and FTK frameworks. In this way, we show that our proposed approach of combining an outlier detection ensemble with a reinforcement learning algorithm improves the accuracy to the extent that even after a single iteration we already accomplish a reduction of 17% in the number of points in need of an evaluation. This shows that there is significant value to be gained in combining outlier detection algorithms with reinforcement learning, especially when the cost to evaluate outliers is high. The adjustment of the actual parameters of the outlier detection, rather than adjusting the ranking of the outliers, this makes it possible to already adjust the outlier detection based on a few instances of labelled data.

There are many applications beyond financial reporting data in which this algorithm can be of use. For example, this approach could be applied to transaction monitoring data. The data volumes are significant and fraudulent transactions are not common and can thus be seen as outliers. Checking whether suspicious transactions are actually fraudulent is time consuming and expensive. Applying the algorithm would over time improve the results of the detection of fraudulent transactions. With each transactions analysed to the point were we can say whether a transaction is truly fraudulent or not, the performance of the outlier detection can be improved.

## Figures and Tables

**Figure 1 entropy-25-00842-f001:**
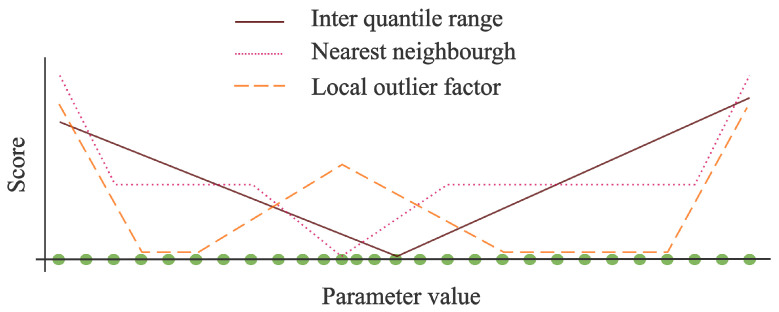
An overview of how the different methods would score points on a one-dimensional scale. With the parameter value the measure of the magnitude of the green datapoints and the score, their respective outlier score for each of the three methods.

**Figure 2 entropy-25-00842-f002:**
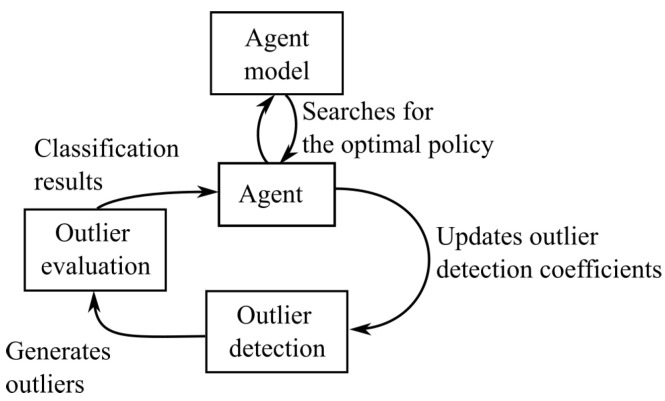
An overview of the reinforcement learning algorithm applied to the outlier detection.

**Figure 3 entropy-25-00842-f003:**
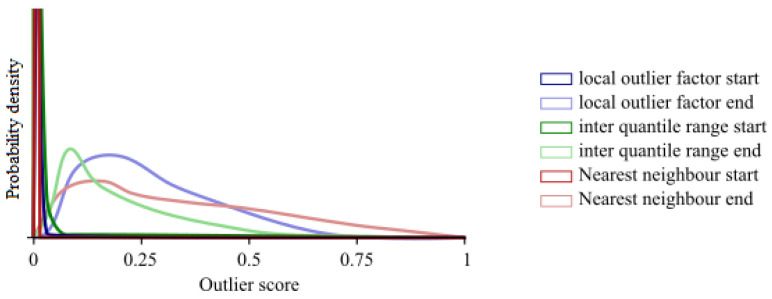
The distribution of the outlier scores for the first and last iteration for the different parts of the ensemble.

**Figure 4 entropy-25-00842-f004:**
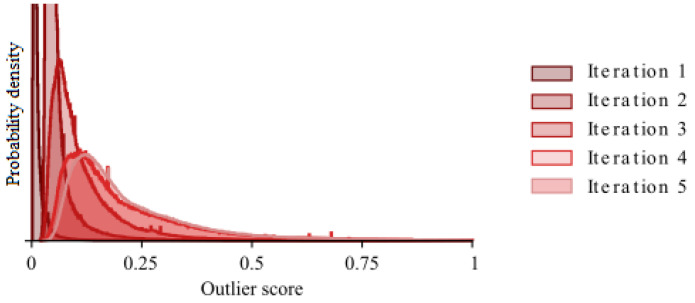
The distribution of the outlier score after each of the iterations.

**Figure 5 entropy-25-00842-f005:**
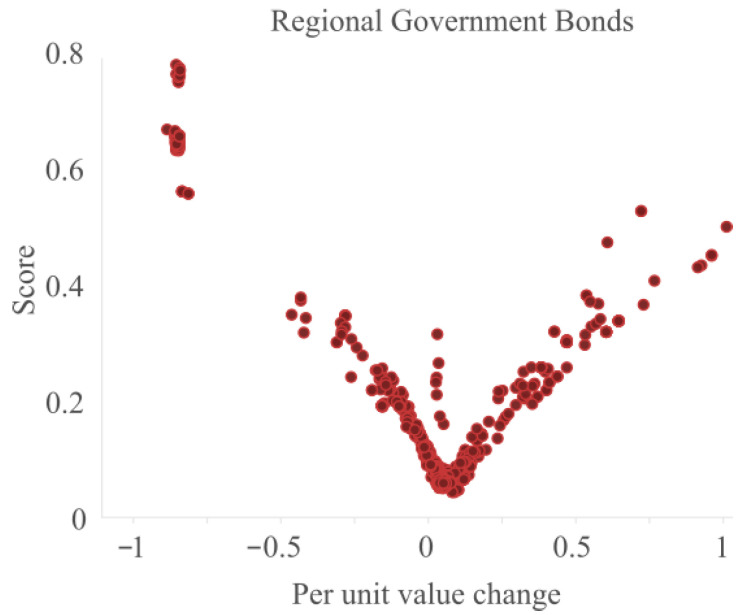
The outlier score versus the per unit change for a subset of the data.

**Figure 6 entropy-25-00842-f006:**
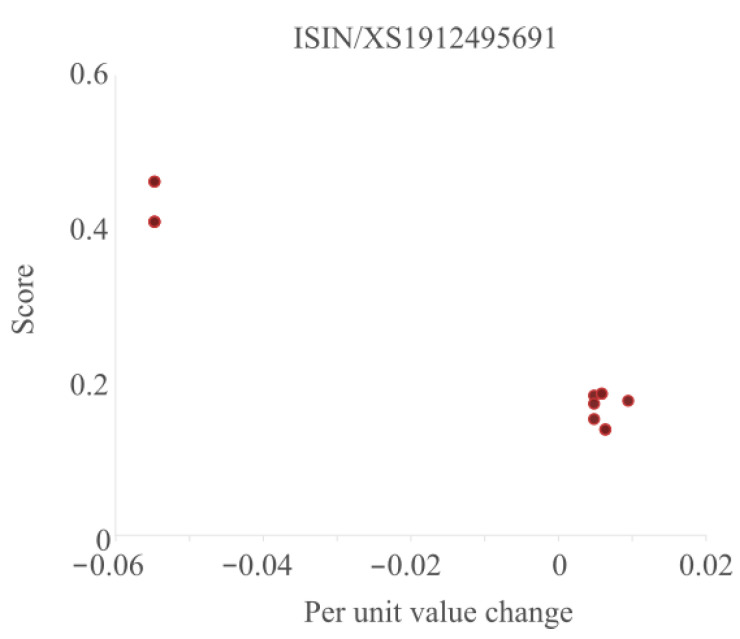
The per unit value versus the score for a single asset.

**Table 1 entropy-25-00842-t001:** The (initial) parameters of the outlier detection as used in this paper.

Parameter	Value	Parameter	Value
$c_{lof}^{k,high}$	0.2	$c_{lof}^{k,low}$	0.2
$c_{iqr}$	0.3	$c_{nn}^{k,high}$	0.15
$k_{nn}^{high}$	50	$c_{nn}^{k,low}$	0.15
$k_{lof}^{high}$	40	$k_{nn}^{low}$	6
$k_{lof}^{low}$	5	$w_{c}$	4.2
$R_{stop}$	0.05	$O_{threshold}$	0.8

**Table 2 entropy-25-00842-t002:** The precision of the ensemble model and other outlier detection algorithms, for the top 500 outliers. LOF (local outlier factor), NN (nearest neighbour), IQR (interquartile range), CBLOF (clustering based local outlier factor), XGBOD (XGBoost based outlier detection).

Ensemble	LOF	NN	IQR	CBLOF	XGBOD
0.967	0.936	0.927	0.899	0.949	0.963

**Table 3 entropy-25-00842-t003:** Descriptive statistics of the per unit value change before and after application of the outlier detection.

	Before	After
Max	13,656,704,633	67.4
99.5th percentile	56.7	3.28
Number of zeroes	4666	2498
0.5th percentile	−10.84	−10.71
Min	−1	−10.95

**Table 4 entropy-25-00842-t004:** The effects of applying the outlier detection algorithm.

	Ensemble Model	
	Initial	Adjusted
Max	6.93	5.60
99.5th percentile	3.20	2.56
median	1.06	1.06
0.5th percentile	−0.59	−0.59
Min	−0.84	−0.84
Outliers vs. non-outliers		
Recall	75.3%	78.6%
Precision	74.8%	82.3%

## Data Availability

Restrictions apply to the availability of these data. Data was obtained from the reporting of Dutch pension funds and insurers to De Nederlandsche Bank. If one wants to perform research on the data, please contact De Nederlandsche Bank at denederlandschebank@dnb.nl.

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
