# Peer review of "Outlier Detection with Reinforcement Learning for Costly to Verify Data"

_entropy, 2023, doi:10.3390/e25060842_

Round 1
Reviewer 1 Report
Expressions (7) and (12) are displayed in a bizarre way. I would preset them differently.
Make sure that the axis of all Figures and the acronyms are fully explained in the figures.
More details on the implementation are necessary. In some places, the authors do not explain exactly how they choose the parameters. At least, the results shoudl bereproducible.
Author Response
Thanks you for taking the time to provide your comments on the manuscript, please see the attachment.

Reviewer 2 Report
The sample heterogeneity, the presence of outliers affect the results of the study. Parametric methods are extremely sensitive to the processing of such data. Therefore, the use of non-parametric methods, in particular machine learning methods - reinforcement learning, is one of the possible approaches to processing such samples, ensuring adequacy and accuracy.
However, with all the relevance of the study, the work has a set of significant shortcomings.
1. Title. Please check if all prepositions are appropriate in your title.
2. Abstract. The phrase "Outliers in data are often present in data" sounds strange and should be rephrased.
3. Figure 1 requires clarification. What is shown on the x-axis?
4. A more detailed explanation is required why this data normalization was chosen, formulas (6), (7).
5. It is not quite clear from the text of the article how the authors design the learning algorithm. That is, it is not clear how the sets of environment states are formed, by what rules the set of agent actions is determined, and how the payoff function is formed.
6. Are there any empirical confirmations of the adequacy of the proposed method. The paper does not provide a comparative analysis of existing approaches and the proposed ensemble method to identifying outliers in data .
7. It is necessary to significantly expand the conclusion, reflecting the difference between the proposed method and others, as well as the practical significance of the study.
8. It is desirable to supplement the references by including sources of practical applications that investigate and overcome the problem of the presence of anomalous values (outliers) in medical, industrial, financial and psychological data
Author Response

(The authors gave the same response as above.)

Round 2
Reviewer 1 Report
No more comments
Reviewer 2 Report
All comments of the reviewer are taken into account.
The manuscript has been sufficiently improved to warrant publication in Entropy.